Boundaries in ground beetle (Coleoptera: Carabidae) and environmental variables at the edges of forest patches with residential developments

Davis Doreen E.
Gagné Sara A. sgagne@uncc.edu
Department of Geography and Earth Sciences, University of North Carolina at Charlotte , Charlotte , NC , United States of America
Ferrenberg Scott
Electronic publication date: 2018 Jan 8
Publication date: 2018
Volume: 6
Electronic Location ID: e4226
Received 2017 Oct 13; Accepted 2017 Dec 13
Copyright: ©2018 Davis and Gagné
Copyright year: 2018
Copyright holder: Davis and Gagné
License: This is an open access article distributed under the terms of the Creative Commons Attribution License, which permits unrestricted use, distribution, reproduction and adaptation in any medium and for any purpose provided that it is properly attributed. For attribution, the original author(s), title, publication source (PeerJ) and either DOI or URL of the article must be cited.
License URL: https://creativecommons.org/licenses/by/4.0/

Keywords: Boundary detection, Edge effects, Temperate deciduous forest, Southeast USA, Urbanization

Funding: University of North Carolina at Charlotte’s Faculty Research Grant program This work was supported by the University of North Carolina at Charlotte’s Faculty Research Grant program. There was no additional external funding received for this study. The funders had no role in study design, data collection and analysis, decision to publish, or preparation of the manuscript.

==============================
Background

Few studies of edge effects on wildlife objectively identify habitat edges or explore non-linear responses. In this paper, we build on ground beetle (Coleoptera: Carabidae) research that has begun to address these domains by using triangulation wombling to identify boundaries in beetle community structure and composition at the edges of forest patches with residential developments. We hypothesized that edges are characterized by boundaries in environmental variables that correspond to marked discontinuities in vegetation structure between maintained yards and forest. We expected environmental boundaries to be associated with beetle boundaries.

Methods

We collected beetles and measured environmental variables in 200 m by 200 m sampling grids centered at the edges of three forest patches, each with a rural, suburban, or urban context, in Charlotte, North Carolina, USA. We identified boundaries within each grid at two spatial scales and tested their significance and overlap using boundary statistics and overlap statistics, respectively. We complemented boundary delineation with k-means clustering.

Results

Boundaries in environmental variables, such as temperature, grass cover, and leaf litter depth, occurred at or near the edges of all three sites, in many cases at both scales. The beetle variables that exhibited the most pronounced boundary structure in relation to edges were total species evenness, generalist abundance, generalist richness, generalist evenness, and Agonum punctiforme abundance. Environmental and beetle boundaries also occurred within forest patches and residential developments, indicating substantial localized spatial variation on either side of edges. Boundaries in beetle and environmental variables that displayed boundary structure at edges significantly overlapped, as did boundaries on either side of edges. The comparison of boundaries and clusters revealed that boundaries formed parts of the borders of patches of similar beetle or environmental condition.

Discussion

We show that edge effects on ground beetle community structure and composition and environmental variation at the intersection of forest patches and residential developments can be described by boundaries and that these boundaries overlap in space. However, our results also highlight the complexity of edge effects in our system: environmental boundaries were located at or near edges whereas beetle boundaries related to edges could be spatially disjunct from them; boundaries incompletely delineated edges such that only parts of edges were well-described by sharp transitions in beetle and/or environmental variables; and the occurrence of boundaries related to edges was apparently influenced by individual property management practices, site-specific characteristics such as development geometry, and spatial scale.

Introduction

Urbanization results in the loss and fragmentation of habitat (Bradley, 1995). Habitat fragments, such as forest patches, in developed landscapes are typically small and irregularly-shaped (Gagné, 2013). Thus, forests in landscapes undergoing urbanization become increasingly dominated by edge habitat, which exhibits markedly altered environmental conditions that may significantly influence the distribution of organisms, such as ground beetles (Coleoptera: Carabidae) (Murcia, 1995).

Ground beetle species richness, abundance, and functional and phylogenetic diversity at the edges of forest patches are higher than (Magura, 2002; Magura, 2017) or similar to (Heliölä, Koivula & Niemelä, 2001; Magura, Lö vei & Tóthmérész, 2017) that in patch interiors. With increasing distance from the edge into a forest patch, the abundance and species richness of forest specialists increase, the abundance and species richness of open-habitat specialists decrease, and the abundances of generalist species show both responses (Gaublomme et al., 2008; Boetzl, Schneider & Krauss, 2016).

Magura (2002) and Gaublomme et al. (2008) have advanced our understanding of forest edge effects on ground beetles by attempting to objectively identify edge and interior habitats and by testing for non-linear responses to distance from the edge, respectively—important approaches that have received relatively little attention by researchers in the field (Ries et al., 2004). Using a two-pane moving window applied to transect data, Magura (2002) defined the edges between forest interior, forest edge, and grass habitats as the locations where community dissimilarity between adjacent sampling locations was at a maximum. Ground beetle species richness differed significantly among the three so-defined habitats. Gaublomme et al. (2008) found that the abundances of individual ground beetle species exhibited steep increases or decreases, accompanied by little change and/or a reversal in the direction of change, in response to distance from the forest edge.

In this paper, we build on these advances by using triangulation wombling to identify boundaries in ground beetle community structure and composition at the developed edges of forest patches. Triangulation wombling is a spatial partitioning technique that identifies locations in sampling grids where variables of interest, such as ground beetle abundance, change dramatically, i.e., boundaries (Fortin & Dale, 2005). Using this method, we sought to address two research questions: (1) what are the spatial patterns of ground beetle community structure and composition and environmental variation at the intersection of forest patches and residential developments; and (2) are the patterns of ground beetle community structure and composition and environmental variation related? We hypothesized that edges between forest patches and residential developments are characterized by boundaries in environmental variables that correspond to marked discontinuities in vegetation structure between maintained yards and forest. Edges between natural and developed land covers are typically characterized by high contrast in vegetation structure (Bennett, 2003). For example, large abrupt changes in the covers of bare ground, grass, leaf litter, and projective foliage occur at the edges of forests and residential developments in Australia (Villaseñor, Blanchard & Lindenmayer, 2016). We expected environmental boundaries to be associated with boundaries in beetle community structure and composition. High contrast edges are hypothesized to exhibit large magnitudes of edge influence due to reduced edge permeability (Ries et al., 2004; Harper et al., 2005). Ground beetles have previously been shown to exhibit sharp changes in community composition at edges (Magura, 2002; Gaublomme et al., 2008; Leslie et al., 2014) and less movement across high contrast than low contrast edges (Collinge & Palmer, 2002).

Materials & Methods

Study area and site selection

We selected three forest patch edges with residential developments in the city of Charlotte, Mecklenburg County, North Carolina, USA, part of one of the most rapidly-growing metropolitan regions in the country (Kotkin & Cox, 2011) (Fig. 1). Hereafter, we use the term “edge” to refer to the property lines between the County-owned forest patches and privately-owned residential developments in our study that approximately corresponded to differences in vegetation management practices. This is in contrast to the term “boundary”, which we use in a technical sense to refer to areas of major change within ground beetle trapping grids. We use the term “site” to refer to the area of each trapping grid.

Figure 1 Map of sites.

The locations of study sites consisting of grids of pitfall traps for ground beetles spanning the edges of forest patches with residential developments in Mecklenburg County, North Carolina, USA. Edges are indicated by black lines and correspond to the property lines between County-owned forest and private development. Each site was at the edge of a patch surrounded by a rural, suburban, or urban context. Aerial imagery courtesy of Mecklenburg County.

We restricted our forest patch selection to upland hardwood forest of as similar a size as possible (0.22–0.63 km2) and similar fractal dimension, an index of patch shape (McGarigal, Cushman & Ene, 2012) (Table 1). All forest patches had a straight, southward-facing edge with a residential development composed of single-family homes (Fig. 1). Each site contained a similar number of houses (Table 1).

Table 1 Site characteristics.

Characteristics of the studied forest patches and the sites encompassing grids of pitfall traps centered at their edges in Charlotte, North Carolina, USA. All patches were upland hardwood forest. Edges were southward-facing and abutted developments with single-family homes.

Patch context (buildings/km2)	Patch area (km2)	Patch fractal dimension	Number of single-family homes in site	Number of pitfall traps in site	Inter-trap distance (m ± SE)	
Rural (<145)	0.63	1.27	9	75	19.8 ± 9.5	
Suburban (145–1,000)	0.22	1.29	13	77	19.5 ± 9.6	
Urban (>1,000)	0.31	1.26	15	78	21.3 ± 10.8	

We also selected forest patches to represent a range of contexts in our study area. For each patch, we quantified intensity of urbanization using a building density raster calculated from 2011 Mecklenburg County tax parcel data and a 1 km-radius moving window, the size of which approximates the dispersal range of ground beetles (Baars, 1979). A rural, suburban, or urban context was attributed to a patch according to the values of the majority of the cells abutting the patch (Table 1).

Ground beetle trapping

Pitfall traps were installed 25 m apart to minimize local depletion of ground beetles (Digweed et al., 1995) in approximately 9 trap × 9 trap grids centered at forest/residential edges, for a total sampling area of 62,500 m2 per site (Fig. 1). Local adjustments to trap location (average inter-trap distance was approximately 20 m) and a reduction in trap number (minimum of 75 traps/site) were necessary to accommodate landowner consent and the presence of impervious surfaces (Table 1).

Each trap consisted of one inner and one outer white cup (Spence & Niemelä, 1994) placed flush with the ground surface, with an opening diameter of 90 mm and a depth of 120 mm. A total of 100 ml of undiluted propylene glycol and a drop of dish soap were used as preservative and surfactant, respectively, and a white roof was placed 25 mm over the trap to prevent the intrusion of rain and debris. We collected trap contents 11 times, approximately every two weeks between April 4 and August 5, 2011. We identified ground beetles to species using Lindroth (1961–1969), Ciegler (2000), and Bousquet (2010). Nomenclature corresponds to that in Bousquet (2012).

Measurement of environmental variables

We measured environmental variables identified as important predictors of ground beetle species richness and abundance (Lövei & Sunderland, 1996; Magura, 2002; Lassau et al., 2005; Magura, Tóthmérész & Elek, 2005) at each trap location. In May 2012, we used a densitometer (Geographic Resource Solutions, Arcata, CA, USA) to estimate the cover of canopy, trees (diameter at breast height (DBH) ≥ 2.5 cm), shrubs (DBH <2.5 cm), woody vines, non-creeping and creeping forbs, grasses, mosses, human-made mulch, coarse woody debris, bare ground, rock, and impervious surfaces. We recorded densitometer readings at 0.92 m intervals along two 3.66 m perpendicular transects centered at each trap location. Transect orientation was determined randomly using a compass. We measured vegetation and ground cover at a height of 1.5 m, with the densitometer pointed upwards for canopy cover and downwards for all other variables. We also recorded leaf litter depth at the same intervals along transects. We estimated the microrelief and slope of the area encompassed by transects by ranking the surface as very even, slightly even, uneven, very uneven, or extremely uneven and of no, weak, moderate, or steep slope, respectively (St-Louis, Fortin & Desrochers, 2004).

In June 2013, we measured surface temperature and relative humidity using LogTag HAXO-8 data loggers (MicroDAQ.com, Contoocook, NH, USA) placed on the ground at each trap location. We recorded data every minute from 12–3 pm on June 21 at the rural site and on June 20 at the suburban site, and from 1–3 pm on June 14 at the urban site. All time periods had similar weather conditions with air temperatures of 25.6–28.3°C, atmospheric pressures of 101.3–102.4 kPa, and partly cloudy skies.

Analyses

We used triangulation wombling to identify boundaries in ground beetle community structure and composition and environmental variation at each site (Fortin & Dale, 2005). In doing so, we assumed that underlying environmental gradients encompassed site areas, a reasonable assumption given the relatively small size of sites. We identified boundaries at two spatial scales, the original sampling scale and a larger scale represented by the centroids of trios of adjacent trap locations (average inter-trap distance was approximately 36 m), to account for the scale-dependency of boundary occurrence (Fortin & Dale, 2005). First, we grouped data points (trap locations at the small scale, centroids at the large scale) into triangles using the Delaunay triangulation algorithm (Fig. S1). Second, we calculated the slope, or rate of change, of a plane fit to the values of a variable of interest observed at each triangle’s vertices using the first partial derivative of the variable in two linear dimensions. We also calculated the orientation, or angle of change, of the plane. Third, rates of change were assigned to the centroids of triangles and were deemed candidate boundary elements if they fell within the top quintile of the distribution of rates of change at the site (Fortin & Dale, 2005). Finally, candidate boundary elements were connected if they met three criteria: they were adjacent to one another; their orientations differed by ≤90°, i.e., they were oriented in roughly the same direction; and the angles between their orientations and the connecting line was ≥30° (so that the connecting line differentiated between areas with very different values of the variable). For simplicity, we refer to connected candidate boundary elements as boundaries and unconnected candidate boundary elements as singletons in the remainder of the paper.

We used BoundarySeer, version 1.5.3 (BioMedware, 2015) to delineate boundaries in the species richnesses, evennesses, and abundances of all beetles, forest specialists, open-habitat specialists, and generalists; the abundance matrix of each of these groups; and the abundances of individual species. We classified species as forest specialists if they were only associated with forest habitats, as open-habitat specialists if they were only associated with open habitats, and as generalists if they were reported to occur in both habitat types according to the accounts in Larochelle & Larivière (2003) (Table S1). Species richnesses were cumulative numbers of species and abundances were sums of individuals collected over the trapping period, divided by the number of successful collections to eliminate the effect of trap disturbance on catch data. We used corrected species richnesses and abundances to calculate species evennesses using the Berger–Parker index (Berger & Parker, 1970), with the exception of forest and open-habitat specialists that were collected at too few traps at each site. Abundance matrices were analyzed in their original form and with species abundances weighted by the inverse of the species’ proportion of total abundance at the site to favor rare species. We also delineated boundaries in standardized environmental variables, both singly and as a matrix.

We determined the significance of the boundary structure exhibited by each variable at each site and scale using boundary statistics (Oden et al., 1993). The significance (p < 0.10) of statistics was assessed by randomizing rates of change 9,999 times assuming complete spatial randomness (Oden et al., 1993). We chose to use a significance threshold greater than 0.05 to lessen Type II error given the novelty of our study objective. Also, Fortin & Dale (2005) suggest that statistics with p > 0.05, in combination with map observations, can be interpreted in order to understand boundary patterns. We also calculated overlap statistics for pairs of beetle and environmental variables, each with one or more significant boundary statistics, in order to assess the degree of spatial coincidence between beetle and environmental boundaries and singletons (Fortin, Drapeau & Jacquez, 1996; Jacquez, 1995). We determined the significance of overlap statistics in the same manner as for boundary statistics but by randomizing only the beetle variable in a pair (BioMedware, 2015). We considered the boundaries and singletons of a beetle-environmental variable pair to exhibit overlap if they occurred significantly close to one another and/or at a significantly large number of the same locations.

Fortin & Dale (2005) recommend complementing boundary detection with clustering to aid in the interpretation of results. We applied the k-means clustering algorithm to variables with one or more significant boundary statistics that displayed boundaries and/or singletons at, near, or parallel to the edge at each site and scale using the cascadeKM function in the vegan 2.4-1 package of R, version 3.3.1 (R Core Team, 2016). We clustered standardized variables into 2–10 groups at the small scale and 2–5 groups at the large scale to explore the spatial coincidence of clusters, boundaries and singletons. We did not cluster single abundance or species richness variables because the Euclidean distance measure used in the k-means algorithm is not appropriate for such data and we were unaware of a transformation to overcome this limitation that is applicable to individual variables.

Figure 2 Boundaries at the rural site.

Boundaries (yellow lines) and singletons (yellow stars) at the rural site in (A) percent grass cover, (B) percent canopy cover, and (C) open-habitat ground beetle abundance (number of individuals per trapping period) at the small scale and in (D) relative humidity, (E) the Berger-Parker index of generalist evenness, and (F) generalist abundance (number of individuals per trapping period) at the large scale. Black dots are trap locations at the small scale and the centroids of trios of adjacent trap locations at the large scale and are labeled with variable values. Edges are indicated by black lines and correspond to the property lines between County-owned forest and private development.

Results

Across all sites, we collected 350 ground beetles per successful trapping period from 26 genera and 50 species (Table S1). The suburban site had the greatest number of individuals per trapping period (132) and species (40), followed by the rural site (115 individuals per trapping period from 31 species) and the urban site (103 individuals per trapping period from 31 species). Most beetle individuals (84%) and species (62%) were generalists. Forest specialists represented 0.6% of individuals and 16% of species whereas open-habitat specialists represented 10% of individuals and 14% of species.

Two-thirds of environmental variables and one quarter of beetle variables had significant boundary statistics at the rural site at the small spatial scale (Table S2, Fig. 2, Fig. S2–S14). Relatively long boundaries in forb, grass (Fig. 2A), and vine cover occurred along part of the edge at the site, overlapping at the back of the central house’s lot. The remainder of environmental variables displayed boundaries in other areas of the site that coincided with major structural change, such as in canopy cover (Fig. 2B). Beetle variables generally exhibited localized spatial variation that did not follow the edge (e.g., Fig. 2C), with the exception of total species evenness, which displayed two short boundaries along the edge.

At the rural site at the large spatial scale, about one third of environmental variables and one third of beetle variables were characterized by significant boundary statistics (Table S3, Fig. 2, Figs. S15–S30). Linear boundaries in temperature and humidity (Fig. 2D) occurred near the edge at the same location that boundaries in forb, grass, and vine covers did at the small scale. Total species evenness and generalist evenness (Fig. 2E) exhibited relatively long linear boundaries parallel to but at a distance into the forest from the edge. Generalist abundance displayed a similar pattern on the developed side of the site (Fig. 2F). All other beetle and environmental variables exhibited localized spatial variation in areas other than at the edge.

Figure 3 Boundaries at the suburban site.

Boundaries (yellow lines) and singletons (yellow stars) at the suburban site in (A) all environmental variables (standardized average of individual variables), (B) percent forb cover, and (C) Poecilus lucublandus lucublandus abundance (number of individuals per trapping period) at the small scale and in (D) P. lucublandus lucublandus abundance (number of individuals per trapping period), (E) leaf litter depth (cm), and (F) total ground beetle abundance (number of individuals per trapping period) at the large scale. Black dots are trap locations at the small scale and the centroids of trios of adjacent trap locations at the large scale and are labeled with variable values. Edges are indicated by black lines and correspond to the property lines between County-owned forest and private development.

One or more boundary statistics was significant for about half of environmental variables and one fifth of beetle variables at the suburban site at the small scale (Table S4, Fig. 3, Figs. S31–S45). The matrix of all environmental variables exhibited a relatively long linear boundary along part of the edge (Fig. 3A) where leaf litter depth, grass cover, shrub cover, and total species evenness also displayed short boundaries. These and the remaining variables displayed boundaries and singletons in other areas of the site as well. In particular, boundaries and singletons in all environmental variables, leaf litter depth, the covers of forb (Fig. 3B), grass, shrub, and creeping forb, total species evenness, generalist evenness, and the abundances of Poecilus lucublandus lucublandus (Fig. 3C) and Pterostichus sculptus occurred along a stream in the forested part of the site.

Boundaries and singletons in P. lucublandus lucublandus abundance (Fig. 3D), as well as in vine cover and leaf litter depth (Fig. 3E), also occurred along the stream at the suburban site at the large scale, at which one fifth of environmental variables and one third of beetle variables had significant boundary statistics (Table S5, Figs. S46–S60). No boundaries in any variable occurred along the edge. Most beetle variables instead displayed very localized spatial variation in other areas of the site (e.g., Fig. 3F).

Eighty-five percent of environmental and 40% of beetle variables had significant boundary statistics at the urban site at the small scale (Table S6, Fig. 4, Figs. S61–S84). Several environmental variables exhibited boundaries along part of the edge of the site, including leaf litter depth (Fig. 4A), slope (Fig. 4B), and the covers of canopy, shrub, and vine. With the exception of leaf litter depth, slope, and shrub cover, these boundaries were restricted to the backyard of the same property. Boundaries in environmental variables, such as canopy cover and impervious surface cover, also delineated structural change in other areas of the site. Several beetle variables, including total abundance and evenness, generalist abundance and evenness, forest specialist abundance and richness, and the abundances of Agonum punctiforme and Scarites subterraneus exhibited a few short boundaries and/or singletons along the edge of the site, mostly at two distinct locations.Of these, the abundance of A. punctiforme displayed the most boundaries and singletons (Fig. 4C). Boundaries and singletons in all beetle variables occurred in other areas of the site as well.

Figure 4 Boundaries at the urban site.

Boundaries (yellow lines) and singletons (yellow stars) at the urban site in (A) leaf litter depth (cm), (B) slope index, and (C) Agonum punctiforme abundance (number of individuals per trapping period) at the small scale and in (D) microrelief index, (E) leaf litter depth (cm), and (F) generalist ground beetle richness (number of species per trapping period) at the large scale. Black dots are trap locations at the small scale and the centroids of trios of adjacent trap locations at the large scale and are labeled with variable values (missing values are indicated by −9,999). Edges are indicated by black lines and correspond to the property lines between County-owned forest and private development.

At the urban site at the large scale, one-third of environmental variables and one-tenth of beetle variables were characterized by significant boundary statistics (Table S7, Fig. 4, Figs. S85–S92). Microrelief, leaf litter depth, and generalist species richness exhibited boundaries along or near parts of the edge (Figs. 4D–4F) and in other areas of the site, as did the remaining variables. One of the boundaries in leaf litter depth was associated with the backyard of the property where several environmental boundaries occurred at the small scale.

Boundaries and singletons in beetle and environmental variables that displayed boundary structure at edges significantly overlapped (Tables S8–S13). At the small scale, boundaries and singletons in total species evenness overlapped those in vine cover at the rural site, those in leaf litter depth and the covers of grass and shrub at the suburban site, and those in slope, leaf litter depth, and vine cover at the urban site. Generalist abundance boundaries and singletons spatially co-occurred with those in temperature at the rural site at the large scale, and generalist evenness boundaries and singletons overlapped those in the covers of canopy, shrub, and vine at the urban site at the small scale. At the urban site at the large scale, boundaries in generalist richness were spatially associated with those in leaf litter depth. Finally, at the urban site at the small scale, boundaries and singletons in forest species abundance and richness significantly overlapped those in leaf litter depth and shrub cover, and boundaries and singletons in the abundances of A. punctiforme and S. subterraneus spatially co-occurred with those in canopy cover.

Many other pairs of beetle and environmental boundaries and singletons exhibited significant spatial overlap (Tables S8–S13). For example, at the rural site at the small scale, boundaries in open-habitat abundance co-occurred with those in temperature, the covers of canopy, forb, and grass, and all environmental variables. At the suburban site at the small scale, boundaries and singletons in the abundances of P. lucublandus lucublandus andP. sculptus overlapped those in creeping forb and shrub covers. P. lucublandus lucublandus boundaries and singletons also overlapped boundaries and singletons in forb and grass covers.

Figure 5 Clusters and boundaries.

Clusters (colored dots), boundaries (yellow lines), and singletons (yellow stars) in grass cover at the rural site at the small scale (A), generalist ground beetle evenness at the rural site at the large scale (B), grass cover (C) and total evenness (D) at the suburban site at the small scale, and leaf litter depth at the urban site at the large scale (E) and at the small scale (F). Edges are indicated by black lines and correspond to the property lines between County-owned forest and private development.

A small number of clusters, typically two or three, coincided with the locations of boundaries and singletons in beetle and environmental variables (Fig. 5, Fig. S93–S108). In general, boundaries and singletons bordered parts of cluster patches, as was the case for the red cluster of grass cover at the rural site at the small scale (Fig. 5A), the blue and white clusters of generalist evenness at the rural site at the large scale (Fig. 5B), the blue and white clusters of grass cover (Fig. 5C) and total species evenness (Fig. 5D) at the suburban site at the small scale, and the blue and white clusters of leaf litter depth at the urban site at the large scale (Fig. 5E). In a few instances, boundaries and singletons corresponded to the borders between patches of some clusters at a site, e.g., the green cluster in leaf litter depth at the urban site at the small scale (Fig. 5F), but not others, e.g., the white cluster at the same site and scale. The spatial patterns of clusters at sites fell into two broad categories, one highlighting the difference between the forested and developed sides of sites (Figs. 5A, 5B, 5E and 5F) and the other the localized, patchy nature of a variable’s distribution throughout a site (Figs. 5C and 5D).

Discussion

Our results demonstrate that the edges of forest patches with residential developments are characterized by boundaries in ground beetle community structure and composition and environmental variation. Boundary structure at edges appeared to be influenced by site-specific factors and individual property management practices. Overlapping boundaries in ground beetle community structure and composition and environmental variation also occurred within forest patches and residential developments, indicating substantial localized spatial variation on either side of edges. These results in combination with those of our clustering analyses suggest that the spatial patterns of ground beetle community structure and composition and environmental variation at the intersection of forest patches and residential developments are described by locations of sharp transition and more gradual gradients.

The beetle variables that exhibited the most pronounced boundary structure in relation to edges were total species evenness, generalist abundance, and generalist evenness at the rural site at the large spatial scale, generalist species richness at the urban site at the large scale, and A. punctiforme abundance at the urban site at the small scale. The long linear boundary in total evenness set back in the forest at the rural site, and in generalist evenness at the same location, likely reflect the spillover of generalist species from the developed side of the site, a pattern that has been demonstrated at grassland edges with cropland (Schneider et al., 2016) and at railway verges bordered by grassland or forest (Prass et al., 2017). Based on the significant overlap of boundaries and singletons in total evenness and vine cover at the rural site at the small scale, we posit that the presence of continuous vine cover in the forest near the edge facilitated the spillover of generalist species, possibly because habitat conditions associated with vine cover were similar to those on the developed side of the site and/or because vine cover posed a significant barrier to the movement of forest specialist species (Magura, 2017).

The location of total and generalist evenness boundaries at the rural site illustrates that major changes in community structure may not occur at the human-defined edge of a patch. Instead, processes associated with the human-defined edge may alter habitat conditions over a wider area, resulting in an organism-perceived ‘edge’ spatially disjunct from the human-perceived one. This spatial incongruity was also apparent in the location of the boundary in generalist abundance on the developed side of the site, which significantly overlapped boundaries in temperature. Temperature is positively associated with generalist beetle species activity (Chiverton, 1988).

Boundaries and singletons in generalist richness and A. punctiforme abundance at the urban site did coincide with the edge of the forest patch. A. punctiforme is a generalist species that is common in disturbed habitats, including those characteristic of residential development (Larochelle & Larivière, 2003). Many short boundaries and singletons in the abundance of the species occurred throughout the developed side of the urban site at the small scale, where they significantly overlapped those in canopy cover, including at the edge. The occurrence of several generalist species nearly exclusively on the developed side of the site, including A. punctiforme and S. subterraneus that also exhibited boundaries and singletons at the edge at the small scale, may underlie the relatively long linear boundary in generalist richness at the edge at the large scale.

Ground beetle and environmental boundary patterns associated with edges appeared to be mediated by individual property management practices, development geometry, and site context. At all three sites, the backyards of single properties abutting edges were the locations of linear boundaries in multiple environmental variables, in some cases at both scales. For example, one property’s backyard at the urban site was delineated by boundaries in slope, leaf litter depth, and the covers of canopy, shrub, and vine at the small scale and leaf litter depth at the large scale. Thus, it seems reasonable to suggest that variation in yard maintenance practices, perhaps associated with local environmental constraints such as steep slopes, may contribute to the location of environmental boundaries at edges. Adding to the influence of this variation is the possible effect of development geometry. The suburban site differed from the other two sites in that roads, and a treed corridor between them, were oriented approximately perpendicular to the forest patch edge rather than parallel to it as at the other sites. This geometry effectively limited the number of backyards abutting the edge and consequently the likelihood of abrupt changes in environmental variation and ground beetle community structure and composition at the edge. Accordingly, the suburban site exhibited the fewest boundaries and singletons associated with the edge at either scale of any site. Finally and notwithstanding the potential effects of management practices and development geometry, we think it possible that site context, in the form of patch area and surrounding building density, may have influenced boundary structure at edges. The rural site was located at the edge of the largest forest patch and exhibited the longest linear boundaries in environmental and ground beetle variables related to the edge at both scales of all three sites. One would expect the strongest edge effects to occur in the largest patch (Soga et al., 2013) surrounded by the least disturbed context (Gaublomme et al., 2008; Jung & Lee, 2016).

Ground beetles exhibited substantial boundary structure that overlapped environmental boundaries and singletons not just at edges, but throughout sites. For example, boundaries and singletons in open-habitat abundance at the rural site at the small scale co-occurred with those in temperature and the covers of canopy, grass, and forb on the developed side of the site. Also, the abundance of P. lucublandus lucublandus displayed boundaries and singletons along the stream in the forested half of the suburban site at the small scale that overlapped boundaries and singletons in the covers of grass, forb, creeping forb, and shrub at the same location. These results highlight not only associations between ground beetle abundance and microclimatic and cover variables that have been reported in the literature (Magura, Tóthmérész & Molnár, 2004; Bergmann et al., 2012), but also illustrate the localized occurrence of ground beetles in response to environmental heterogeneity.

The superposition of boundaries, singletons, and clusters revealed that boundaries and singletons formed parts of the borders of patches of similar ground beetle or environmental condition, including those that differentiated between the forested and developed sides of sites. The fact that boundaries and singletons only bordered parts of patches demonstrates that edge effects at our sites manifested as abrupt transition in some places, indicated by the presence of boundaries and singletons, and more gradual change in others, indicated by their absence. The existence of this variation in the form of edge effects at our sites is not surprising given the high degree of spatial heterogeneity characteristic of urban areas. Our comparison of the results of boundary and cluster analyses supports the idea that the approaches together describe the locations and shapes of patches in an area and the form of borders among them, as also demonstrated by Fortin & Drapeau (1995).

Conclusions

We show that edge effects on ground beetle community structure and composition and environmental variation at the intersection of forest patches and residential developments can be described by boundaries and that these boundaries often overlap in space. We found relatively long boundaries in multiple environmental variables, such as grass cover and leaf litter depth, and beetle variables, such as total species evenness, associated with edges. However, our results also highlight the complexity of edge effects in our system: environmental boundaries were located at or near edges whereas beetle boundaries related to edges could be spatially disjunct from them; boundaries incompletely delineated edges such that only parts of edges were well-described by sharp transitions in beetle and/or environmental variables; and the occurrence of boundaries related to edges was apparently influenced by individual property management practices, site-specific characteristics such as development geometry, and spatial scale. Also, boundaries occurred throughout site areas, revealing substantial localized spatial variation in environmental conditions and ground beetle community structure and composition within 100 m either side of edges. We hope that our identification of boundaries and the complexity that it revealed are fodder for future investigations of organismal patterns at heterogeneous urban edges.

Supplemental Information

Supplemental Information 1 Table S1. Species habitat associations and abundances

Habitat associations and abundances (total numbers of individuals per trap collection) of ground beetle species found at study sites.

Click here for additional data file.

Supplemental Information 2 Tables S2–S7. Boundary statistics

Boundary statistics calculated for environmental and ground beetle variables. Beetle community matrices were analyzed using raw abundances and species abundances weighted by the inverse of the species’ proportion of total abundance at the site. NS: total number of boundaries and singletons; N1: number of singletons; Lmean: mean length of boundaries and singletons (number of candidate boundary elements); Lmax: maximum length; Dmean: mean diameter of boundaries (minimum number of links between the farthest pair of candidate boundary elements); Dmax: maximum diameter; D∕L: mean diameter-to-length ratio. ∗p < 0.10.

Click here for additional data file.

Supplemental Information 3 Tables S8–S13. Overlap statistics

Overlap statistics calculated for pairs of environmental and ground beetle variables. Shown for each pair: Oe, the mean minimum Euclidean distance between each environmental candidate boundary element and the closest beetle candidate boundary element (m); Ob, the mean minimum Euclidean distance between each beetle candidate boundary element and the closest environmental candidate boundary element (m); Oeb, the overall mean minimum Euclidean distance between beetle and environmental candidate boundary elements (m); and Os, the number of beetle and environmental candidate boundary elements at the same locations. † Significantly low value (p < 0.10). ∗ Significantly high value (p < 0.10).

Click here for additional data file.

Supplemental Information 4 Figure S1. Delaunay triangles and their centroids

The Delaunay triangles (black lines) and their centroids (white diamonds) at each site at two spatial scales. Black dots are trap locations at the small scale and the centroids of trios of adjacent trap locations at the large scale. Edges are indicated by dashed black lines and correspond to the property lines between County-owned forest and private development.

Click here for additional data file.

Supplemental Information 5 Figures S2–S92. Boundary figures

Boundaries (yellow lines) and singletons (yellow stars) in environmental and ground beetle variables with one or more significant boundary statistics. Figures are named by site (rural, suburban, or urban), spatial scale (small or large), and variable. Black dots are trap locations at the small scale and the centroids of trios of adjacent trap locations at the large scale and are labeled with variable values (temperature in degrees Celsius; humidity as a percentage; microrelief and slope as ordinal indices; leaf litter depth in cm; covers as percentages; all environmental variables as a standardized average of individual variables; abundances and richnesses as numbers of individuals and species, respectively, per trapping period; evennesses as Berger-Parker indices; missing values indicated by −9,999). Edges are indicated by black lines and correspond to the property lines between County-owned forest and private development. Beetle community matrices were analyzed using raw abundances and species abundances weighted by the inverse of the species’ proportion of total abundance at the site.

Click here for additional data file.

Supplemental Information 6 Figures S93–S108. Clusters and boundaries

Clusters (colored dots), boundaries (yellow lines), and singletons (yellow stars) in environmental and ground beetle variables that exhibited boundaries and singletons at, near, or parallel to edges. Figures are named by site (rural, suburban, or urban), spatial scale (small or large), and variable. Edges are indicated by black lines and correspond to the property lines between County-owned forest and private development.

Click here for additional data file.

Data S1 Raw data

Data used in the delineation of boundaries. Missing values are indicated by −9,999. Species abundances are numbers of individuals per trapping period.

Click here for additional data file.

We very much thank Sandra Clinton, William Garcia, and two anonymous reviewers for their helpful feedback on an earlier version of this manuscript. We are also indebted to Luke Browder, Rufus McLean, and Jacob Todd who helped to collect field data, to Kelly Brawn for help with all things GIS, and to Janet Ciegler who confirmed our beetle identifications. Very special thanks are due to the landowners at our sites, including the Mecklenburg County Division of Nature Preserves and Natural Resources, whose cooperation made this research possible.

Additional Information and Declarations

Competing Interests

Author Contributions

Field Study Permissions

Data Availability

The authors declare there are no competing interests.

Doreen E. Davis performed the experiments, analyzed the data, wrote the paper, prepared figures and/or tables, reviewed drafts of the paper.

Sara A. Gagné conceived and designed the experiments, analyzed the data, wrote the paper, prepared figures and/or tables, reviewed drafts of the paper.

The following information was supplied relating to field study approvals (i.e., approving body and any reference numbers):

We received a permit to collect ground beetles from the Mecklenburg County Division of Nature Preserves and Natural Resources.

The following information was supplied regarding data availability:

The raw data has been provided as a Supplemental File.

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
