# Peer review of "Boundaries in ground beetle (Coleoptera: Carabidae) and environmental variables at the edges of forest patches with residential developments"

_PeerJ, doi:10.7717/peerj.4226_

## Round 0.1 · original submission · Minor Revisions

Given the substantial changes made in this manuscript compared to the previously reviewed versions, it was my decision to request additional expert reviews. Both reviewers see this version as an improvement and have made suggestions for minor revisions. I agree with the reviewers that with some relatively minor changes this manuscript will be ready for publication.

In particular, take note of literature recommended by Reviewer 1 that can be incorporated into the introduction and discussion to capture the most timely research in this area. Also, Reviewer 2 poses some questions regarding key definitions and aspects of the analytical approaches to identifying boundaries. Addressing these points with simple text additions in the methods will help to reduce any similar questions or confusion for readers.

In addition to these points, I recommend some minor text edits detailed below. Please note that PeerJ does not offer suggestions for text editing at the proof stage so be sure to thoroughly proof your manuscript ahead of submitting a revision.

Line 112: Please note the units in "9 x 9" as either "9 m x 9 m" or "9 x 9 m square". Also, using the multiplication symbol instead of the letter x is recommended.

Line 114: Please state the lowest number of traps used in any site compared to the 230 commonly deployed.

Line 120-121: Please state the total number of times traps were collected so that readers do not have to calculate this value from the dates.

Line 124: This one sentence long paragraph can be combined with the subsequent paragraph of a similar theme.

Line 136: Again, this one sentence paragraph can be combined with the previous paragraph on environmental characterization.

Lines 177-187: This section can be one paragraph.

Reviewer 1 ·

Basic reporting

No comment.

Experimental design

No comment.

Validity of the findings

No comment.

Additional comments

This interesting manuscript is dealing with the identification of boundaries in environmental variation and ground beetle community structure using triangulation wombling. Moreover, the authors tried to determine whether these boundaries are related. The topic of the manuscript is very interesting and relevant, since ecological responses to the presence of habitat edges are one of the most extensively researched topics in ecology. The main strength of this manuscript is the use of a novel and powerful spatial partitioning technique, the triangulation wombling, to identify boundaries. Objectively identifying boundaries is a key topic in edge studies, as different groups of organisms may operate at different spatial scales, and what appears as a homogenous patch to one species may comprise a very heterogeneous patchy environment to another. Earlier this manuscript was submitted to the Ecosphere and the Ecology and Evolution and was thoroughly reviewed. Based on the reviewers’ comments, the authors completely re-wrote and substantially improved the manuscript. The present form of the manuscript is generally well written. The analyses are well explained and straightforward, and the discussion is sound. The most recent edge studies on ground beetles (e.g. Urban Ecosystems, 20: 971-981, 2017; Forest Ecology and Management, 384: 371-377, 2017; Journal of Insect Conservation, 20: 49-57, 2016; Ecological Research, 31: 799-810, 2016), however, are not referred in the manuscript. Please cite and discuss these papers. Based on the above, this manuscript is valuable and should be published.

Reviewer 2 ·

Basic reporting

I am not an expert on beetles, pit traps, nor triangulation-based methods, so I can only comment from a general science and statistics perspective.

I don't know the "triangulation wombling" was explained well for a journal of general readership. E.g.:
"Second, we calculated a rate of change and its orientation for each triangle using the values of a variable of interest at the triangle’s vertices."
Rate of change of what with respect to what? How do you define the "orientation" of a triangle?
I had to look up the referenced book to determine that "traingulation wombling" involves a finite-difference method for approximating spatial derivatives in multiple dimensions and orientation is that of the approximated gradient.
This method naturally would involve some assumptions about continuity that I doubt are checked in practice, but for detecting "boundaries" that run perpendicular to large gradients perhaps that doesn't matter.

From a statistical perspective, the layers of estimation atop the data were a bit uncomfortable.
First, finite-difference derivatives are calculated.
From that boundaries are detected at p=10% significance and with various thresholds.
Then the boundary overlaps were calculated, also at p=10% significance.
This looks like a typical use of this method, but I do not find it ideal.

Otherwise, the paper looks fine to me.

Experimental design

no comment

Validity of the findings

no comment

Additional comments

no comment

External reviews were received for this submission. These reviews were used by the Editor when they made their decision, and can be downloaded below.

---

## Round 0.2 · accepted · Accept

The new version of this manuscript sufficiently incorporated reviewer suggestions and addressed all editorial concerns. The authors also accounted for reviewer comments from a prior review at a different journal. The identification of habitat boundaries and transitions is increasingly important in ecology and conservation. I agree with the authors that this shorter exploration and validation of a novel method, submitted to PeerJ, represents an advance. The methods tested and described in this manuscript have great potential to guide advanced efforts to more quantitatively identify boundaries in both biotic and abiotic variables. Thank you for this fine contribution.

External reviews were received for this submission. These reviews were used by the Editor when they made their decision, and can be downloaded below.